# Neural dynamics of visual working memory representation during sensory distraction

**Jonas Karolis Degutis[1,2,3]\*, Simon Weber[1,4], Joram Soch[1,5,6,7], John-Dylan Haynes[1,2,3,4,8,9]**

[1]Bernstein Center for Computational Neuroscience Berlin and Berlin Center for Advanced Neuroimaging, Charité Universitätsmedizin Berlin, Corporate member of the Freie Universität Berlin, Humboldt-Universität zu Berlin, and Berlin Institute of Health, Berlin, Germany; [2]Max Planck School of Cognition, Munich, Germany; [3]Department of Psychology, Humboldt-Universität zu Berlin, Berlin, Germany; [4]Research Training Group 'Extrospection' and Berlin School of Mind and Brain, Humboldt-Universität zu Berlin, Berlin, Germany; [5]Institute of Psychology, Otto von Guericke University, Magdeburg, Germany; [6]Max Planck Institute for Human Cognitive and Brain Sciences, Leipzig, Germany; [7]German Center for Neurodegenerative Diseases, Bonn, Germany; [8]Collaborative Research Center 'Volition and Cognitive Control', Technische Universität Dresden, Dresden, Germany; [9]Research Cluster of Excellence 'Science of Intelligence', Technische Universität Berlin, Berlin, Germany

**\*For correspondence:**
j.karolis.degutis@
maxplanckschools.de

**Competing interest:** The authors declare that no competing interests exist.

## eLife Assessment

This **important** study reports a reanalysis of one experiment of a previously-published report to characterize the dynamics of neural population codes during visual working memory in the presence of distracting information. This paper presents **solid** evidence that working memory representations are dynamic and distinct from sensory representations of intervening distractions. This research will be of interest to cognitive neuroscientists working on the neural bases of visual perception and memory.

**Abstract** Recent studies have provided evidence for the concurrent encoding of sensory percepts and visual working memory (VWM) contents across visual areas; however, it has remained unclear how these two types of representations are concurrently present. Here, we reanalyzed an open-access fMRI dataset where participants memorized a sensory stimulus while simultaneously being presented with sensory distractors. First, we found that the VWM code in several visual regions did not fully generalize between different time points, suggesting a dynamic code. A more detailed analysis revealed that this was due to shifts in coding spaces across time. Second, we collapsed neural signals across time to assess the degree of interference between VWM contents and sensory distractors, specifically by testing the alignment of their encoding spaces. We find that VWM and feature-matching sensory distractors are encoded in coding spaces that do not fully overlap, but the separation decreases when distractors negatively impact behavioral performance in recalling the target. Together, these results indicate a role of dynamic coding and temporally stable coding spaces in helping multiplex perception and VWM within visual areas.

## Introduction

To successfully achieve behavioral goals, humans rely on the ability to remember, update, and ignore information. Visual working memory (VWM) allows for a brief maintenance of visual stimuli that are no longer present within the environment (*Curtis and D'Esposito, 2003*; *Goldman Rakic, 1995*; *D'Esposito and Postle, 2015*). Previous studies have revealed that the contents of VWM are present throughout multiple visual areas, starting from V1 (*Fuster and Alexander, 1971*; *Harrison and Tong, 2009*; *Christophel et al., 2012*; *Ester et al., 2015*; *Rademaker et al., 2019*; *Rademaker et al., 2019*; *Riggall and Postle, 2012*; *Serences et al., 2009*; *Christophel et al., 2017*; *Curtis and Sprague, 2021*). These findings raised the question of how areas that are primarily involved in visual perception can also maintain VWM information without interference between the two contents. Recent studies that had participants remember a stimulus while simultaneously being presented with sensory stimuli during the delay period have found supporting evidence that both VWM contents and sensory percepts are multiplexed in occipital and parietal regions (*Rademaker et al., 2019*; *Iamshchinina et al., 2021a*; *Bettencourt and Xu, 2016*). However, the mechanism employed to segregate bottom-up visual input from VWM contents remains poorly understood.

One proposed mechanism to achieve the separation between sensory and memory representations is dynamic coding (*Stokes et al., 2020*; *Stokes, 2015*; *Stroud et al., 2024*): the change of the population code encoding VWM representations across time. Recent work has shown that the format of VWM might not be as persistent and stable throughout the delay as previously thought (*Miller et al., 2018*; *Sreenivasan et al., 2014a*). Frontal regions display dynamic population coding across the delay during the maintenance of category (*Meyers et al., 2008*) and spatial contents in the absence of interference (*Murray et al., 2017*; *Spaak et al., 2017*), and also show dynamic recoding of the memoranda after sensory distraction (*Parthasarathy et al., 2017*; *Parthasarathy et al., 2019*). The visual cortex in humans displays dynamic coding of contents during high load trials (*Sreenivasan et al., 2014b*) and during a spatial VWM task (*Li and Curtis, 2023*). However, it is not yet clear whether dynamic coding of VWM might help evade sensory distraction in human visual areas.

Another line of evidence suggests that perception could potentially be segregated from VWM representations using stable nonoverlapping coding spaces (*Lorenc et al., 2021*). For example, evidence from neuroanatomy indicates that the sensory bottom-up visual pathway primarily projects to the cytoarchitectonic Layer 4 in V1, while feedback projections culminate in superficial and deep layers of the cortex (*Felleman and Van Essen, 1991*). Functional results are in line with neuroanatomy by showing that VWM signals preferentially activate the superficial and deep layers in humans (*Lawrence et al., 2018*) and nonhuman primates (*van Kerkoerle et al., 2017*), while perceptual signals are more prevalent in the middle layers (*Lawrence et al., 2019*). In addition to laminar separation, regional multiplexing of multiple items could potentially rely on rotated representations, as seen in memory and sensory representations orthogonally coded in the auditory cortex (*Libby and Buschman, 2021*) and in the storage of a sequence of multiple spatial locations in the prefrontal cortex (PFC) (*Xie et al., 2022*). Nonoverlapping orthogonal representations have also been seen in both humans and trained recurrent neural networks as a way of segregating attended and unattended VWM representations (*Wan et al., 2022*; *Wan et al., 2023*; *van Loon et al., 2018*).

Here, we investigated whether the concurrent presence of VWM and sensory information is compatible with predictions offered by dynamic coding or by stable nonaligned coding spaces. For this, we reanalyzed an open-access fMRI dataset by *Rademaker et al., 2019*, where participants performed a delayed-estimation VWM task with and without sensory distraction. To investigate dynamic coding, we employed a temporal cross-decoding analysis that assessed how well the multivariate code encoding VWM generalizes from one time point to another (*Spaak et al., 2017*; *Stokes et al., 2013*; *Degutis et al., 2023*; *Anders et al., 2011*), and a temporal neural subspace analysis that examined a sensitive way of looking at alignment of neural populations coding for VWM at different time points. To assess the nonoverlapping coding hypothesis, we used neural subspaces (*Murray et al., 2017*; *Li and Curtis, 2023*; *Libby and Buschman, 2021*) to see whether temporally stable representations of the VWM target and the sensory distractor are coded in separable neural populations. Finally, we examined the multivariate VWM code changes during distractor trials when compared to the no-distractor VWM format.

## Results

### Temporal cross-decoding in distractor and no-distractor trials

In the previously published study (*Rademaker et al., 2019*). two groups of participants completed two VWM experiments where on a given trial they were asked to remember an orientation of a grating, which they had to then recall at the end of the trial. In the first experiment, the delay period was either left blank (no-distractor) or a noise or randomly oriented grating distractor was presented (*Figure 1a*). To investigate the dynamics of the VWM code, we examined how the multivariate pattern of activity encoding VWM memoranda changed across the duration of the delay period. To do so, we ran a temporal cross-decoding analysis where we trained a decoder (periodic support vector regression [pSVR], see *Weber et al., 2024*) on the target orientation, separately for each time point and tested on all time points in turn in a cross-validated fashion. If the information encoding VWM memoranda were to have the same code, the trained decoder would generalize to other time points, indicated by similar decoding accuracies on the diagonal and off-diagonal elements of the matrix. However, if the code exhibited dynamic properties, despite information about the memoranda being present (above-chance decoding on the diagonal of the matrix), both off-diagonal elements corresponding to a given on-diagonal element would have lower decoding accuracies (*Figure 1b*). Such off-diagonal elements are considered an indication of a dynamic code.

We ran the temporal cross-decoding analysis in the first of two experiments for the three VWM delay conditions: no-distractor, noise distractor, and orientation distractor (feature-matching distractor). First, we examined each element of the cross-decoding matrix to test whether decoding accuracies were above chance. In all three conditions and throughout all regions of interest (ROIs), we found clusters where decoding was above chance (*Figure 1c–e*, black outline; nonparametric cluster permutation test against null; all clusters $p<0.05$) from as early as 4 s after the onset of the delay period. We found that decoding on the diagonal was highest during no-distractor compared to noise and orientation distractor trials in most ROIs (Figure 4a).

Second, we examined off-diagonal elements to assess whether there was any indication that they reflected a non-generalizing dynamic code (see Methods for full details). Despite a high degree of temporal generalization, we found dynamic coding clusters in all three conditions. Some degree of dynamic coding was observed in all ROIs but LO2 in the noise distractor and no-distractor trials, while it was only present in V1, V2, V3, V4, and IPS in the orientation distractor condition (*Figure 1c–e*, blue outline). The difference between noise and orientation distractor conditions could not be explained by the amount of information present in each ROI, as the decoding accuracy of the diagonal was similar across all ROIs in both the noise and orientation distractor conditions (Figure 4a). We saw a nominally larger number of dynamic coding elements in V1, V2, and V3AB during the noise distractor condition and in V3 during the no-distractor condition (*Figure 1*).

To qualitatively compare the amount of dynamic coding in the three conditions across the delay period, we calculated a dynamicism index (*Spaak et al., 2017*; *Figure 1e*; see Methods), which measured the multivariate code's uniqueness at each time point; more precisely, the proportion of dynamic elements corresponding to each diagonal element. High values indicate dynamic code, and low values indicate a generalizing code. Across all conditions, most dynamic elements occurred between the encoding and early delay periods (4–8 s), and the late delay and retrieval (14.4–16.8 s). Interestingly, during the noise distractor trials in V1, we also saw dynamic coding during the middle of the delay period; the multivariate code not only changed during the onset and offset of the noise stimulus, but also during its presentation and throughout the extent of the delay.

We also ran the same temporal cross-decoding analysis on the second experiment from *Rademaker et al., 2019*. Participants performed the same type of VWM task with three conditions: a blank delay (no-distractor), a naturalistic distractor, and a flickering orientation distractor (*Figure 1—figure supplement 1a*). The decoding accuracies across all conditions, including the no-distractor condition, were nominally lower in the second experiment compared to the first. In some ROIs, the information about the target was not present during the delay period (chance decoding on the diagonal; *Figure 1—figure supplement 1b–d*). Similarly to the first experiment's orientation trials, the flickering orientation condition did not exhibit high levels of dynamic coding of the target (*Figure 1—figure supplement 1b–d*). Due to the decreased amount of target information present across time points in the two distractor conditions (*Figure 1—figure supplement 1e*), we do not further investigate the temporal dynamics of VWM storage in the second experiment.

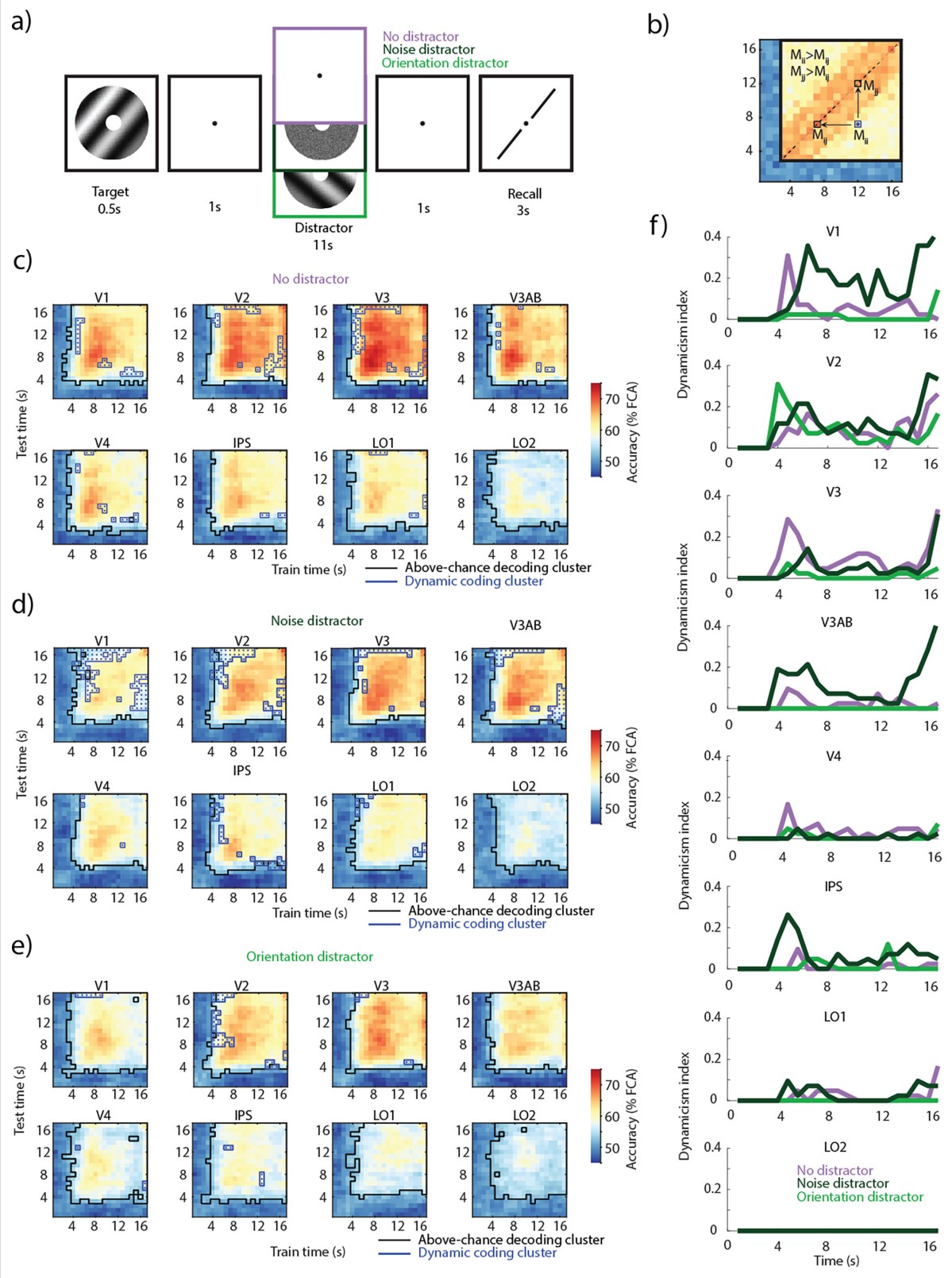

**Figure 1.** Task and temporal cross-decoding. (**a**) On each trial, an oriented grating was presented for 0.5 s followed by a delay period of 13 s (*Rademaker et al., 2019*). In a third of the trials, a noise distractor was presented for 11 s during the middle of the delay; in another third, another orientation grating was presented; one-third of trials had no-distractor during the delay. (**b**) Illustration of dynamic coding elements. An off-diagonal element had to have a lower decoding accuracy compared to both corresponding diagonal elements (see Methods for details). (**c**) Temporal

*Figure 1 continued on next page*

*Figure 1 continued*

generalization of the multivariate code encoding VWM representations in three conditions across occipital and parietal regions. Across-participant mean temporal cross-decoding of no-distractor trials. Black outlines: matrix elements showing above-chance decoding (cluster-based permutation test; p<0.05). Blue outlines with dots: dynamic coding elements; parts of the cross-decoding matrix where the multivariate code fails to generalize (off-diagonal elements having lower decoding accuracy than their corresponding two diagonal elements; conjunction between two cluster-based permutation tests; p<0.05). (**d**) Same as **c**, but noise distractor trials. (**e**) Same as **c**, but orientation distractor trials. (**f**) Dynamicism index; the proportion of dynamic coding elements across time. High values indicate a dynamic non-generalizing code, while low values indicate a generalizing code. Time indicates the time elapsed since the onset of the delay period.

The online version of this article includes the following figure supplement(s) for figure 1:

**Figure supplement 1.** Task and temporal cross-decoding of Experiment 2.

**Figure supplement 2.** Simulations.

**Figure supplement 3.** Selected voxels.

We conducted several simulations to investigate whether changes in signal-to-noise ratio (SNR) could lead to dynamic coding clusters in our temporal cross-decoding analyses. This could occur if a decoder trained at one SNR fails to generalize to another, because each may rely on different features within the data. Specifically, we varied the added noise level in either a simulated dataset of voxel responses or the empirical results from V1 in the no-distractor or noise distractor trials in the first experiment. We then trained on a given noise level and tested on all other noise levels, corresponding to the temporal cross-decoding analysis. We see an absence of dynamic elements in the SNR cross-decoding matrix, as the decoding accuracy more strongly depends on the training data rather than test data. This results in some off-diagonal values in the decoding matrix that are higher, rather than smaller, than corresponding on-diagonal elements (*Figure 1—figure supplement 2a*). In the second and third simulations, we used empirical data. To follow the initial decrease and subsequent increase in decoding accuracy found in most ROIs (Figure 4a), we initially decreased and then increased the SNR in the train and test axes of the matrix (*Figure 1—figure supplement 2*). Similarly to the first simulation, the cross-decoding matrix lacked dynamic elements, with the decoding accuracy of most off-diagonal elements exceeding their corresponding on-diagonal elements (*Figure 1—figure supplement 2b and c*). These simulations indicate that SNR differences are unlikely to give rise to dynamic coding clusters.

## Dynamics of VWM neural subspaces across time

The temporal cross-decoding analysis of the first experiment revealed more dynamic coding in the early visual cortex primarily during the early and late delay phase and more generalized coding throughout the delay in higher-order regions. To understand the nature of these effects in more detail, we conducted a separate series of analyses that directly assessed the neural subspaces in which the orientations were encoded and how these potentially changed across time. Specifically, we followed a previous methodological framework (*Li and Curtis, 2023*) and applied a principal component analysis (PCA) to the high-dimensional activity patterns at each time point to identify the two axes that explained maximal variance across orientations (see *Figure 2* and Methods).

First, we visualized the consistency of the neural subspaces across time. For this, we computed low-dimensional 2D neural subspaces for a given time point and projected left-out data from six time points during the delay onto this subspace (*Li and Curtis, 2023*; *Libby and Buschman, 2021*). A projection of data from a single time point resulted in four orientation bin values placed within the subspace (*Figure 2a*, colored circles indicate orientation). Considering projected data from all time points, if the VWM code were generalizing, we would see a clustering of orientation points in a subspace; however, if orientation points were scattered around the neural subspace, this would show a non-generalizing code.

We examined the projections in a combined ROI spanning V1-V3AB aggregated across participants. We projected left-out data from all six time point bins onto subspaces generated from the early (7.2 s), middle (12 s), and late (16.8 s) time point data for each of the three conditions. Overall, the results showed generalization across time with some exceptions (*Figure 2c*, *Figure 2—figure supplement 1*). The clustering of orientation bins in the no-distractor condition was most pronounced (Figure 4a). In contrast, the noise distractor trials showed a resemblance of some degree of dynamic coding, as seen by less variance explained by early time points projected onto the middle subspace

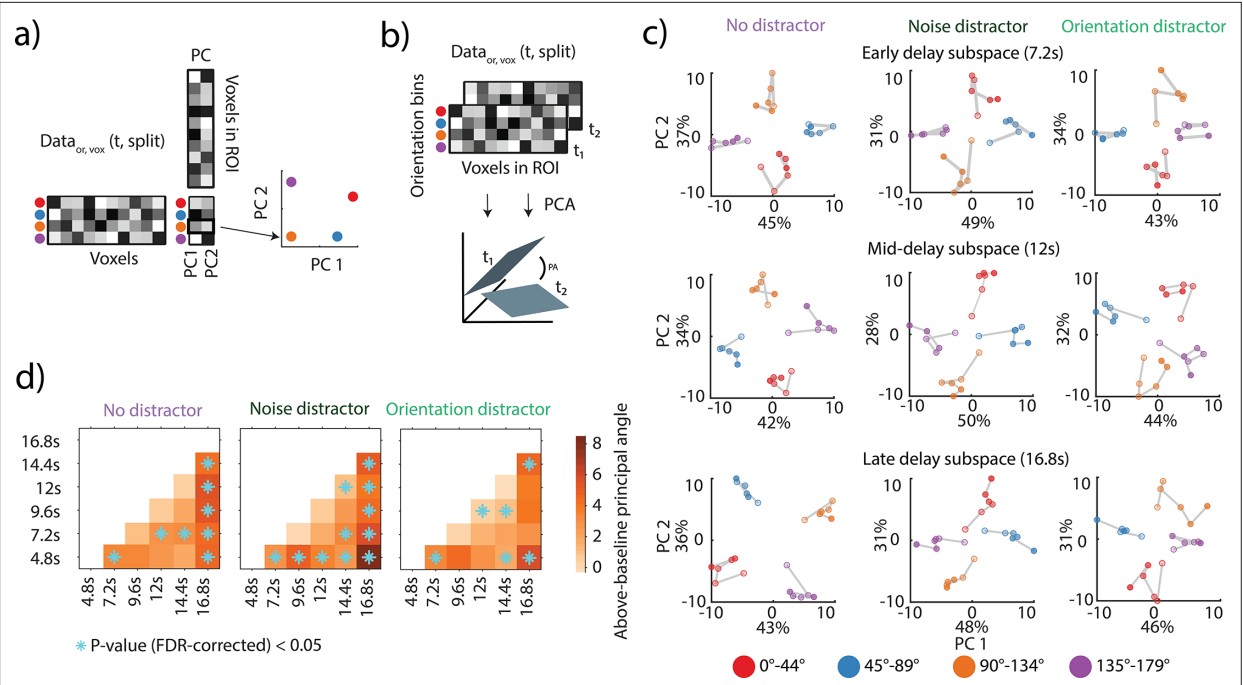

**Figure 2.** Assessing the dynamics of neural subspaces in V1-V3AB. (**a**) Schematic illustration of the neural subspace analysis. A given data matrix (voxels × orientation bins) was subjected to a principal components analysis (PCA), and the first two dimensions were used to define a neural subspace onto which a left-out test data matrix was projected. This resulted in a matrix of two coordinates for each orientation bin and was visualized (see right). The *x* and *y* axes indicate the first two principal components. Each color depicts an angular bin. (**b**) Schematic illustration of the calculation of an above-baseline principal angle (aPA). A principal angle (PA) is the angle between the 2D PCA-based neural subspaces (as in **a**) for two different time points $t_1$, $t_2$. A small angle would indicate alignment of coding spaces; an angle of above-baseline would indicate a shift in the coding space. The aPA is the angle for a comparison between two time points ($t_1$, $t_2$) minus the angle between cross-validated pairs of the same time points. (**c**) Each row shows a projection that was estimated for one of two time ranges (middle and late delay) and then applied to all time points (using independent, split-half cross-validated data). Opacity increases from early to late time points. For visualization purposes, the subspaces were estimated on a participant-aggregated region of interest (ROI) (*Li and Curtis, 2023*). The axes represent the first two principal components, with labels indicating the percent of total explained variance. *Figure 2—figure supplement 1* depicts the same projections as neural trajectories. (**d**) aPA between all pairwise time point comparisons (nonparametric permutation test against null; FDR-corrected p<0.05) averaged across 1000 split-half iterations. Corresponding p-values can be found in *Supplementary file 1, table S1*.

The online version of this article includes the following figure supplement(s) for figure 2:

**Figure supplement 1.** Neural trajectories across time.

and the early and middle time points projected onto the late subspace (*Figure 2*, *Figure 2—figure supplement 1*).

To quantify the visualized changes, we measured the alignment between each pair of subspaces by calculating the above-baseline principal angle (aPA) (*Figure 2b*) within the combined V1-V3AB ROI. The aPA measures the alignment between the 2D subspaces encoding the VWM representations: the higher the angle, the smaller the alignment between two subspaces and an indication of a changed neural coding space. Unlike in the projection of data from time points, the aPA was calculated participant-wise. Using a split-half approach, we measured the aPA between each split-pair of subspaces and subtracted the angles measured within each of the subspaces, with the latter acting as a null baseline.

All three conditions showed significant aPAs (*Figure 2d*; cyan stars; permutation test; p<0.05, FDR-corrected). Corresponding to the results from the cross-decoding analysis, the early (4.8 s) and late (16.8 s) delay subspaces showed the highest number of significant pairwise aPAs in all conditions, with noise distractor trials having all pairwise aPAs, including the early and late subspaces, being significant. The three conditions each had two significant aPAs between time points in the middle of the delay period.

## Alignment between distractor and target subspaces in orientation distractor trials

Next, we assessed any similarity in encoding between the memorized orientation targets and the orientation distractors by focusing on those trials where both occurred. First, we examined whether the encoding of the sensory distractor in the first experiment is stable across its entire presentation duration (1.5–12.5 s after target onset) using the same approach as for the VWM target (*Figure 1e*). We found stable coding of the distractor in all ROIs with only a few dynamic elements in V2 and V3 (*Figure 3—figure supplement 1*). We then assessed whether the sensory distractor had a similar code to the VWM target by examining whether the multivariate code across time generalizes from the target to the distractor and vice versa. When cross-decoded, the sensory distractor (*Figure 3— figure supplement 1c*) and target orientation (*Figure 3—figure supplement 1b*) had lower decoding accuracies in the early visual cortex compared to when trained and tested on the same label type,

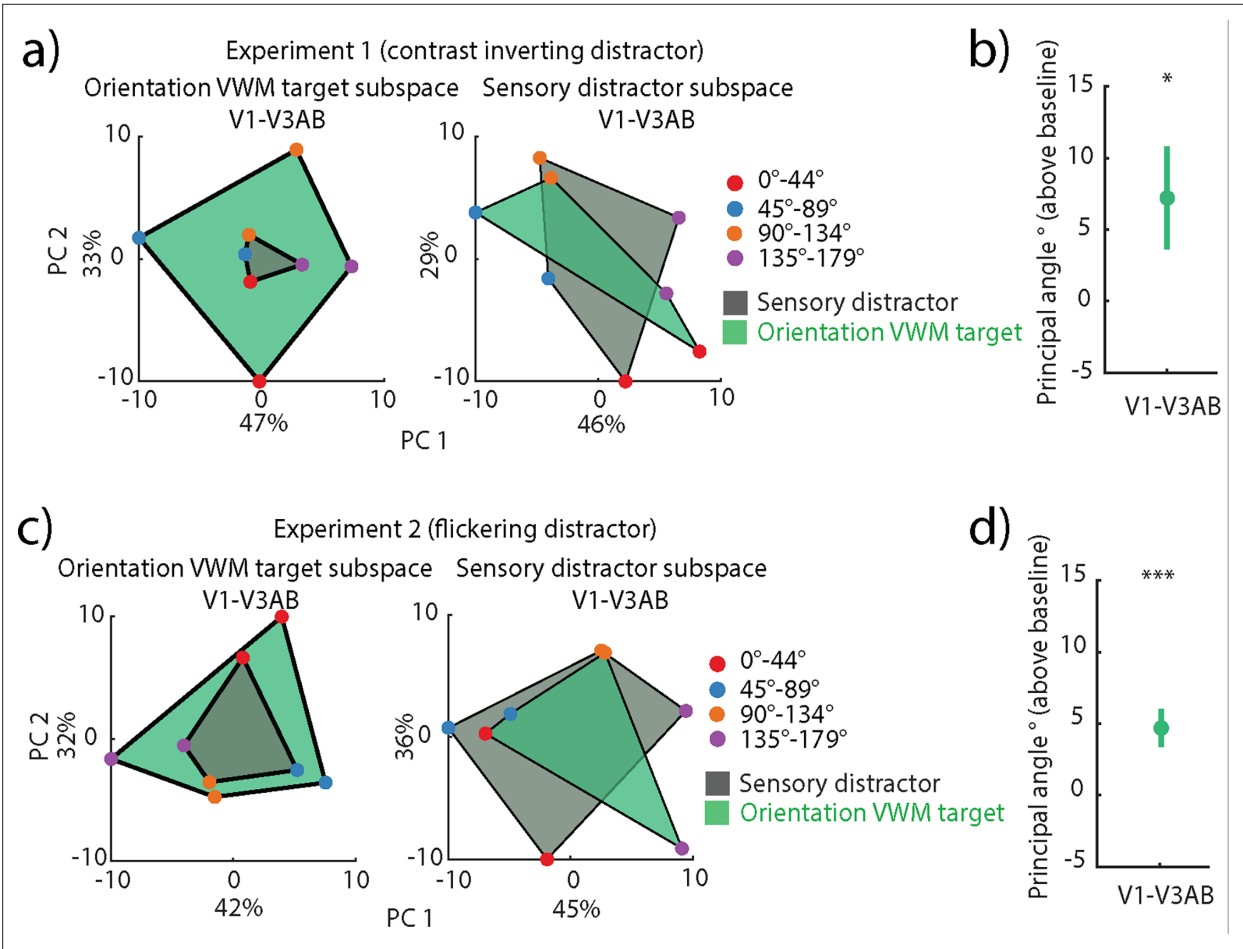

**Figure 3.** Generalization between target and distractor codes in orientation distractor visual working memory (VWM) trials in V1-V3AB. (**a**) Left: projection of left-out target (green) and sensory distractor (gray) onto an orientation VWM target neural subspace. Right: same as left, but the projections are onto the sensory distractor subspace. The axes represent the first two principal components, with labels indicating the percent of total explained variance. (**b**) Principal angle between the sensory distractor and orientation VWM target subspaces (p=0.0297, one-tailed permutation test of sample mean), averaged across 1000 split-half iterations. Error bars indicate SEM across participants. (**c**) Same as **a**, but for flickering orientation distractor trials in the second experiment. (**d**) Same as **b**, but for flickering orientation distractor trials in the second experiment (p<0.001, one-tailed permutation test of sample mean). The same figure for individual regions of interest (ROIs) can be seen in *Figure 3—figure supplement 3*.

The online version of this article includes the following figure supplement(s) for figure 3:

**Figure supplement 1.** Temporal cross-decoding of distractor and memory target in orientation distractor trials in Experiment 1.

**Figure supplement 2.** Temporal cross-decoding of distractor and memory target in flickering orientation distractor trials in Experiment 2.

**Figure supplement 3.** Stable coding spaces of memory target and distractor in each region of interest (ROI).

indicative of a non-generalizing code. Such a difference was not seen in higher-order visual regions, as the decoding of the sensory distractor was low to begin with (*Figure 3—figure supplement 1a–c*).

Since we found minimal dynamics in the encoding of the distractor (*Figure 3—figure supplement 1a*) and target (*Figure 1e*) in the first experiment, we focused on temporally stable neural subspaces that encoded the target and sensory distractor. We computed stable neural subspaces where we disregarded the temporal variance by averaging across the whole delay period and binned the trials based on either the target orientation (*Figure 3a*, left subpanel) or the distractor orientation (*Figure 3a*, right subpanel). We then projected left-out data binned based on the target (*Figure 3a*, green quadrilateral) or the distractor (*Figure 3a*, gray quadrilateral). This projection provided us with both a baseline (as when training and testing on the same label) and a cross-generalization. Unsurprisingly, the target subspace explained the left-out target data well (*Figure 3a*, left subpanel, green quadrilateral); however, the target subspace explained less variance of the left-out distractor data (*Figure 3a*, left subpanel, gray quadrilateral), as qualitatively seen from the smaller spread of the sensory distractor orientations. A similar but less pronounced dissociation between projections was seen in the distractor subspace (*Figure 3a*, left, quadrilateral in green) with the distractor subspace better explaining the left-out distractor data. We quantified the difference between the target and distractor subspaces and found a significant aPA between them (p=0.0297, one-tailed nonparametric permutation test; *Figure 3b*).

We ran the same analyses on the flickering orientation distractor trials in the second experiment, in which the flickering distractor trials negatively impacted target recall compared to the no-distractor trials (*Rademaker et al., 2019*). Similarly to the first experiment, we find reliable temporal cross-decoding of the distractor (*Figure 3—figure supplement 2a*) and a non-generalizing code when trained and tested on the target and distractor, respectively, and vice versa (*Figure 3—figure supplement 2b and c*). As the target (*Figure 1—figure supplement 1c*) and distractor (*Figure 3—figure supplement 2a*) displayed limited coding dynamics, we also computed stable neural subspaces of the target and distractor. As in the first experiment, the target subspace explained the left-out target data well (*Figure 3b*, left subpanel, green quadrilateral); however, the same subspace explained the left-out distractor better than in the first experiment (*Figure 3a*, left subpanel, gray quadrilateral), as seen from the larger overlap between the projected distractor and target quadrilaterals. Similarly, there was a large overlap between the projections in the distractor subspace (*Figure 3b*, right subpanel). The aPA was significant (p<0.001, one-tailed nonparametric permutation test; *Figure 3d*), but nominally smaller than in the first experiment. The results from the first and second experiments provide evidence for the presence of separable stable neural subspaces that might enable the multiplexing of VWM and perception across the extent of the delay period, and the separation of these subspaces is impacted by behavioral performance.

## Impact of distractors on VWM multivariate code

To further assess the impact of distractors on the available VWM information, we examined the decoding accuracies of distractor and no-distractor trials across time in the first experiment. Decoding accuracy was higher in the no-distractor trials compared to both orientation and noise distractor trials across all ROIs, but IPS (*Figure 4a*, red and blue lines, p<0.05, cluster permutation test) across several stages of the delay period. To further assess how distractors affected the delay period information, we increased sensitivity by collapsing across it, because time courses were comparable in all conditions (*Figure 4a*). To assess to which degree VWM encoding generalized from no-distractor to distractor trials, we trained a decoder on no-distractor trials and tested it on both types of distractor trials (*Figure 4b* noise- and orientation-cross). We expressed the decoding accuracy of each distractor condition as a proportion of the decoding accuracy in the no-distractor condition. Values close to one indicate comparable information, while values below one mean the decoder does not generalize well. We found that the cross-decoding accuracies were significantly lower than the no-distractor in all ROIs but V4 (in both noise and orientation) and LO2 (only noise). Thus, in most areas, the decoder did not generalize well from the no-distractor to distractor conditions. However, the total amount of information in distractor trials was generally slightly lower (*Figure 4a*). Thus, we also compared the generalization to a decoder trained and tested on the same distractor condition (*Figure 4b* noise- and orientation-within), which might thus be able to extract more information. We found that indeed information was recovered in areas V2 and V3AB in the noise distractor condition (*Figure 4b*, pairwise

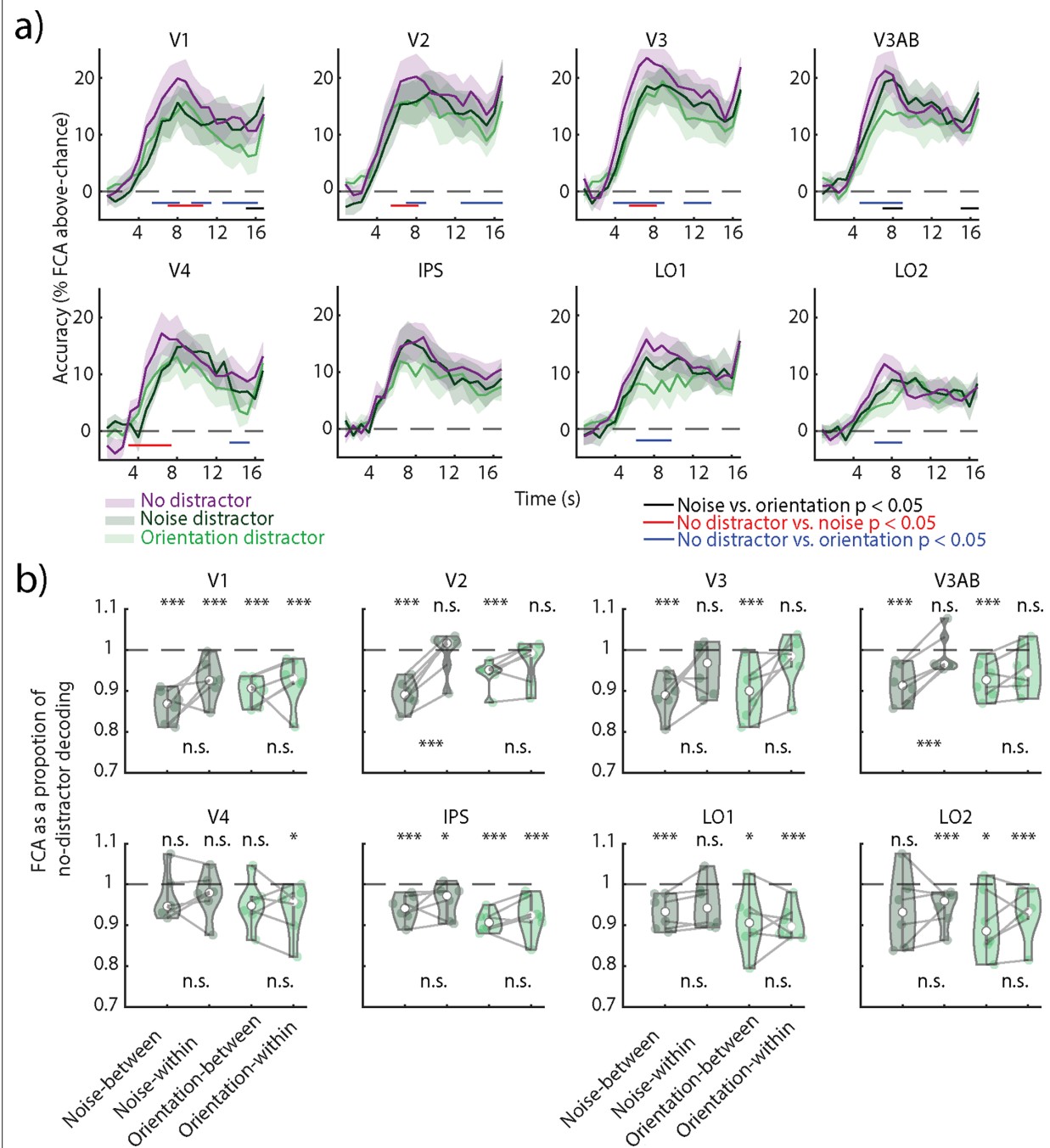

**Figure 4.** Cross-decoding between distractor and no-distractor conditions in Experiment 1. (**a**) Decoding accuracy (feature continuous accuracy [FCA]) across time for train and test on no-distractor trials (purple), train and test on noise distractor trials (dark green), and train and test on orientation distractor trials (light green). Horizontal lines indicate clusters where there is a difference between two time courses (all clusters p<0.05; nonparametric cluster permutation test, see color code on the right). (**b**) Decoding accuracy as a proportion of no-distractor decoding estimated on the averaged delay period (4–16.8 s). Nonparametric permutation tests compared the decoding accuracy of each analysis to the no-distractor decoding baseline (indicated as a dashed line) and between a decoder trained and tested on distractor trials (noise- or orientation-within) and a decoder trained on no-distractor trials and tested on distractor trials (noise- or orientation-cross). FDR-corrected across regions of interest (ROIs). *p<0.05, ***p<0.001. Corresponding p-values can be found in ***Supplementary file 1, table S2***.

The online version of this article includes the following figure supplement(s) for figure 4:

**Figure supplement 1.** Temporal cross-decoding generalization between distractor and no-distractor visual working memory (VWM) trials.

**Figure supplement 2.** Cross-decoding between distractor and no-distractor conditions in Experiment 2.

permutation test). Thus, there was more information in the noise distractor condition, but it was not accessible to a decoder trained only on no-distractor trials. Additionally, a temporal cross-decoding analysis where all training time points were no-distractor trials had less dynamic coding in early visual regions (*Figure 4—figure supplement 1*) when compared to the temporal cross-decoding matrix when trained and tested on noise distractor trials (*Figure 1d*). These results indicate a change in the VWM format between the noise distractor and no-distractor trials.

We find a similar pattern of results in the second experiment's naturalistic distractor condition, where there is a recovery of information in V3 and LO2 and LO1 in the flickering orientation distractor condition (*Figure 4—figure supplement 2*).

## Discussion

We examined the dynamics of VWM with and without distractors and explored the impact of sensory distractors on the coding spaces of VWM contents in visual areas by reanalyzing previously published data (*Rademaker et al., 2019*). In two experiments, participants completed a task during which they had to maintain an orientation stimulus in VWM. In the first experiment, the delay period either had no-distractor, an orientation distractor, or a noise distractor. The second experiment had either no-distractor, flickering orientation distractor, or naturalistic distractor trials. We assessed two potential mechanisms that could help concurrently maintain the superimposed sensory and memory representations. First, we examined whether changes were observable in the multivariate code for memory contents across time, which we term dynamic coding. For this, we used two different analyses: temporal cross-classification and a direct assessment of angles between coding spaces. We found evidence for dynamic coding in all conditions of the first experiment, but there were differences in these dynamics between conditions and regions. Dynamic coding was most pronounced during the noise distractor trials in early visual regions. Second, we assessed the complementary question of temporally stable coding spaces. We computed the stable neural subspaces by averaging across the delay period and saw that coding of the VWM target and concurrent sensory distractors occurred in different stable neural subspaces. This overlap was less pronounced in the second experiment where the flickering orientation distractor impacted behavioral performance when recalling the memory target (*Rademaker et al., 2019*). Finally, we observed that the format of the multivariate VWM code during the noise distraction differs from the VWM code when distractors were not present.

Dynamic encoding of VWM contents has been repeatedly examined before. Temporal cross-decoding analyses have been used in a number of nonhuman primate electrophysiology and human fMRI studies (*Stokes et al., 2020*; *Spaak et al., 2017*; *Parthasarathy et al., 2017*; *Sreenivasan et al., 2014b*; *Li and Curtis, 2023*; *Degutis et al., 2023*; *Cavanagh et al., 2018*). *Spaak et al., 2017*, found dynamic coding in the nonhuman primate PFC during a spatial VWM task. They observed a change in the multivariate code between different stages; specifically, a first shift between the encoding and maintenance periods, and also a second shift between the maintenance and retrieval periods. The initial transformation between the encoding and maintenance periods might recode the percept of the target into a stable VWM representation, whereas the second might transform the stable memoranda into a representation suited for initiation of motor output. A similar dynamic coding pattern was also observed in human visual regions using neuroimaging (*Li and Curtis, 2023*). In this study, in all three conditions, we find a comparable pattern of results, where the multivariate code changes between the early delay and middle delay, and middle delay and late delay periods.

When noise distractors are added to the delay period, we find evidence of additional coding shifts in V1 during the middle of the delay. Previous research in nonhuman primates has shown that the presentation of a distractor induces a change in multivariate encoding for VWM in lateral PFC (lPFC) (*Parthasarathy et al., 2017*). More precisely, a lack of generalization was observed between the population code encoding VWM before the presentation of a distractor (first half of the delay) and after its presentation (the second half). Additionally, continuous shifts in encoding have been observed in the extrastriate cortex throughout the extent of the delay period when decoding multiple remembered items at high VWM load (*Sreenivasan et al., 2014b*). The dynamic code has been interpreted to enable multiplexing of representations when the visual cortex is overloaded by the maintenance of multiple stimuli at once. Future research could examine how properties of the distractor and of the target stimulus could interact to lead to dynamic coding. One intriguing hypothesis is that distractors that perturb the activity of feature channels that are used to encode VWM representations

induce changes in its coding space over time. It is important to note that in the first experiment, the activation of the encoded target features was highest for the noise stimulus. Thus, the shared spatial frequencies between the noise distractor and the VWM contents potentially contribute to a more pronounced dynamic coding effect.

In a complementary analysis, we directly assessed subspaces in which orientations were encoded in VWM. We defined the subspaces for three different time windows: early, middle, and late. We find no evidence that the identity of orientations is confusable across time, e.g., we do not observe 45° at one given time point being recoded as 90° from a different time point. Such dynamics have been previously observed in the rotation of projected angles within a fixed neural subspace (*Libby and Buschman, 2021*; *Wan et al., 2022*) Rather, we find a decreased generalization between neural subspaces at different time points, as previously observed in a spatial VWM task (*Li and Curtis, 2023*). These results suggest that the temporal dynamics across the VWM trial periods are driven by changes in the coding subspace of VWM. We do observe a preservation of the topology of the projected angles, as more similar angles remained closer together (e.g. the bin containing 45° was always closer to the bin containing 0° and 90°). Such a topology has been seen in V4 during a color perception task (*Brouwer and Heeger, 2009*).

We also find evidence that the VWM contents are encoded in a different way depending on whether a noise distractor is presented or not. The decoder trained on no-distractor trials does not generalize well, presumably because it fails to fully access all the information present in noise distractor trials. If the decoders are trained directly on the distractor conditions, the VWM-related information is much higher. Additionally, we see that the code generalizes better across time when training on no-distractor trial time points and testing on noise distractor trials. This may imply that by training our decoder on the no-distractor trials, we are able to uncover an underlying stable population code encoding VWM in noise distractor trials. Consistent with this finding, *Murray et al., 2017*, demonstrated that subspaces derived on the delay period could still generalize to the more dynamic encoding and retrieval periods, albeit not perfectly.

Interestingly, we found limited dynamic coding in the orientation distractor condition; primarily a change in the code between the early delay and middle delay periods was observed. Nonetheless, we find distinct temporally stable coding spaces in which sensory distractors and memory targets are encoded. These results correspond to prior research demonstrating a rotated format between perception and memory representations (*Libby and Buschman, 2021*), attended and unattended VWM representations in both humans and recurrent-neural networks trained on a 2-back VWM (*Wan et al., 2022*) and serial retro-cueing tasks (*Wan et al., 2023*; *Piwek et al., 2023*). Additionally, similar rotational dynamics have been observed between multiple spatial VWM locations stored in the nonhuman primate lPFC (*Xie et al., 2022*). Considering the consistency of these results across different paradigms, we speculate that separate coding spaces might be a general mechanism of how feature-matching items can be concurrently multiplexed within visual regions. With growing evidence of the relationship between VWM capacity and neural resources available within the visual cortex (*Cohen et al., 2014*; *Franconeri et al., 2013*; *Sprague et al., 2014*), further research could examine the number of feature-matching items that can be stored in nonaligned coding spaces.

In this study, the first experiment did not yield a behavioral deficit in the feature-matching orientation distractor trials, while the second experiment impacted target recall in the feature-matching flickering orientation trials (*Rademaker et al., 2019*). There is evidence from behavioral and neural studies that show interactions between perception and VWM: feature-matching distractors behaviorally bias retrieved VWM contents (*Rademaker et al., 2015*; *Mallett et al., 2020*); VWM representations influence perception (*Teng and Kravitz, 2019*; *Kang et al., 2011*; *Gayet et al., 2013*; *Gayet et al., 2017*); neural visual VWM representations in the early visual cortices are biased toward distractors (*Lorenc et al., 2018*); and the fidelity of VWM neural representations within the visual cortex negatively correlates with behavioral errors when recalling VWM during a sensory distraction task (*Hallenbeck et al., 2021*). In cases where a distractor does induce a drop in recall accuracy or biases the recalled VWM target, VWM and the sensory distractor neural subspaces might overlap more, as we see in the second experiment. However, it remains to be seen whether the degree of change or rotation between subspaces correlates with trial-to-trial behavior.

To our surprise, we did not observe a significant difference in the coding format of VWM between orientation distractor and no-distractor trials. Our initial expectation was that the VWM coding might

undergo changes due to the target representation avoiding the distractor stimulus. However, the presence of a generalizing code between no-distractor and orientation distractor trials both in the first and second experiments, along with the nonaligned coding spaces between the target and distractor in both orientation trials, suggests an alternative explanation. We suggest that the sensory distractor stimulus occupies a distinct coding space throughout its presentation during the delay, while the coding space of the target remains the same in both orientation and no-distractor trials. Layer-specific coding differences in perception and VWM might explain these findings (*Lawrence et al., 2018*; *Lawrence et al., 2019*; *Iamshchinina et al., 2021b*). Specifically, the sensory distractor neural subspace might predominantly reside in the bottom-up middle layers of early visual cortices, while the neural subspace encoding VWM might primarily occupy the superficial and deep layers.

We provide evidence for two types of mechanisms found in visual areas during the presence of both VWM and sensory distractors. First, our findings show dynamic coding of VWM within the human visual cortex during sensory distraction and indicate that such activity is present not only within the lPFC. Second, we find that VWM and feature-matching sensory distractors are encoded in shifted coding spaces, but the overlap between these subspaces increases in trials that negatively affect recall fidelity. Considering previous findings, we posit that different coding spaces within the same region might be a more general mechanism of segregating feature-matching stimuli. In sum, these results provide possible mechanisms of how VWM and perception are concurrently present within visual areas.

## Methods
### Participants, stimuli, procedure, and preprocessing
The following section is a brief explanation of parts of the methods covered in *Rademaker et al., 2019*. Readers may refer to that paper for details. We reanalyzed data from Experiments 1 and 2.

In Experiment 1, six participants performed two tasks while in the scanner: a VWM task and a perceptual localizer task. In the perceptual localizer task, either a donut-shaped or a circle-shaped grating was presented in 9 s blocks. The participants had to respond whenever the grating dimmed. There was a total of 20 donut-shaped and 20 circle-shaped gratings in one run. Participants completed a total of 15–17 runs.

The visual VWM task began with the presentation of a colored 100% valid cue, which indicated the type of trial: no-distractor, orientation distractor, or noise distractor. Following the cue, the target orientation grating was presented centrally for 500 ms, followed by a 13 s delay period. In the trials with the distractor, a stimulus of the same shape and size as the target grating was presented centrally for 11 s in the middle of the delay period (*Figure 1a*). The orientation and noise distractors reversed contrast at 4 Hz. At the end of the delay, a probe stimulus bar appeared at a random orientation. The participants had to align the bar to the target orientation and had to respond in 3 s. The orientations for the VWM sample were pseudorandomly chosen from six orientation bins, each consisting of 30 orientations. The orientation distractor and sample were counterbalanced in order not to have the same orientation presented as a distractor. Each run consisted of four trials of each condition. Across three sessions, participants completed 27 runs of the task, resulting in a total of 108 trials per condition.

In Experiment 2, seven participants performed the same type of VWM task with three trial types: no-distractor, flickering orientation distractor, and a naturalistic (gazebo or face) distractor (*Figure 1— figure supplement 1a*). The memory target presentation was the same as in Experiment 1. However, unlike Experiment 1, where distractors exhibited contrast reversals, the distractors in Experiment 2 alternated between being displayed on and off at 4 Hz. In the naturalistic distractor condition, a full set of 22 unique face images or 22 unique gazebo images was presented in a randomly shuffled sequence. In the flickering grating distractor condition, 22 gratings were shown, each sharing the same orientation but with a randomly assigned phase. As in Experiment 1, the orientations of both the target and distractor gratings were pseudorandomly drawn from six predefined orientation bins.

The data were acquired using a simultaneous multi-slice EPI sequence with a TR of 800 ms, TE of 35 ms, flip angle of 52°, and isotropic voxels of 2 mm. The data were preprocessed using FreeSurfer and FSL, and time series were z-scored across time for each voxel.

## Voxel selection

We used the same ROIs as in *Rademaker et al., 2019*, which were derived using retinotopic mapping. In contrast to the original study, we reduced the size of our ROIs by selecting voxels that reliably responded to both the donut-shaped orientation perception task and the no-distractor VWM task. To select reliably activating voxels, we calculated four tuning functions for each voxel: two from the perceptual localizer and two from the no-distractor VWM task. The tuning functions spanned the continuous feature space in bins of 30°. Thus, to calculate the tuning functions, we ran a split-half analysis using stratified sampling where we binned all trials into six bins (of 30°). For both halves, tuning functions were estimated using a GLM that included six orientation regressors (one for each bin) and assumed an additive noise component independent and identically distributed across trials. We calculated Pearson's correlations between the no-distractor memory and the perception tuning functions across the six parameter estimates extracted from the GLM, thus generating one memory-memory and one perception-perception correlation coefficient for each voxel.

The same analysis was additionally performed 1000 times on randomly permuted orientation labels to generate a null distribution for each participant and each ROI. These distributions were used to check for the reliability of voxel activation to perception and no-distractor VWM. After performing Fisher's z-transformation on the correlations, we selected voxels that had a value above the 75th percentile of the null distributions in both the memory-memory and perception-perception correlations. This population of voxels was then used for all subsequent analyses. IPS included reliable voxels from retinotopically derived IPS0, IPS1, and IPS2. The number and proportion of voxels selected for each experiment is seen in *Figure 1—figure supplement 3*.

## Periodic support vector regression

We used pSVR to predict the target orientation from the multivariate BOLD activity (*Weber et al., 2024*). PSVR uses a regression approach to estimate the sine and cosine components of a given orientation independently and therefore accounts for the circular nature of stimuli. To have a proper periodic function, orientation labels from the range [0°, 180°) were projected into the range [0, $2\pi$].

We used the support vector regression algorithm using a nonlinear radial basis function kernel implemented in LIBSVM (*Chang and Lin, 2011*) for orientation decoding. Specifically, sine and cosine components of the presented orientations were predicted based on multivariate fMRI signals from a set of voxels at specific time points within a trial (see *Temporal cross-decoding*). In each cross-validation fold, we rescaled the training data voxel activation into the range [0, 1] and applied the training data parameters to rescale the test data. For each participant, we had a total of three iterations in our cross-validation, where we trained on two-thirds (i.e. two sessions) and tested on one-third of the data (i.e. the left-out session). We selected three iterations to mitigate training and test data leakage (see *Temporal cross-decoding*).

After pSVR-based analysis, reconstructed orientations were obtained by plugging the predicted sine and cosine components into the four-quadrant inverse tangent:

$$\theta_p = atan2(x_p, y_p)$$

where $x_p$ and $y_p$ are pSVR outputs in the test set. Prediction accuracy was measured as the trial-wise absolute angular deviation between predicted orientation and actual orientation:

$$\Delta x = \left| (\theta - \theta_p)_{\text{circ}} \right|$$

where $\theta$ is the labeled orientation and $\theta_p$ is the predicted orientation. This measure was then transformed into a trial-wise feature continuous accuracy (FCA) (*Pilly and Seitz, 2009*) as follows:

$$\text{FCA} = \frac{\pi - \Delta x}{\pi} \cdot 100$$

The final across-trial accuracy was the mean of the trial-wise FCAs. Mean FCA was calculated across predicted orientations from all test sets after cross-validation was complete. The FCA is an equivalent measurement to standard accuracy measured in decoding analyses falling into the range between 0% and 100% but extended to the continuous domain. In the case of random guessing, the expected angular deviation is $\pi/2$, resulting in chance-level FCA at 50%.

## Temporal cross-decoding

To determine the underlying stability of the VWM code, we ran a temporal cross-decoding analysis using pSVR (*Figure 1*). We trained on data from a given time point and then predicted orientations for all time points, using the presented targets as labels. We trained on two-thirds of the trials per iteration and tested on the left-out third. Training and test data were never taken from the same trials, both when testing on the same and different time points.

We used a cluster-based approach to test for significance for above-chance decoding clusters (*Maris and Oostenveld, 2007*) To determine whether the size of the cluster of the above-chance values was significantly larger than chance, we calculated a summed t-value for each cluster. We then generated a null distribution by randomly permuting the sign of the estimated above-chance accuracy (each FCA value was subtracted by 50%, such that 0 corresponds to chance level) of all components within the temporal cross-decoding matrix. We calculated the summed t-value for the largest randomly occurring above-chance cluster. This procedure was repeated 1000 times to estimate a null distribution. The empirical summed t-value of each cluster was then compared to the null distribution to determine significance (p<0.05; without control of multiple cluster comparisons).

Dynamic coding clusters were defined as elements within the temporal cross-decoding matrix, where the multivariate code at a given time point did not fully generalize to another time point; in other words, an off-diagonal element was significantly smaller in accuracy compared to its two corresponding on-diagonal elements ($a_{ij} < a_{ii}$ and $a_{ij} < a_{jj}$, *Figure 1b*). In order to test for the significance of these clusters, we ran two cluster permutation tests as done in previous studies to define dynamic clusters (*Spaak et al., 2017*; *Li and Curtis, 2023*). In each test, we subtracted one or the other corresponding diagonal elements from the off-diagonal elements ($a_{ij} - a_{ii}$ and $a_{ij} - a_{jj}$). We then ran the same sign permutation test as for the above-chance decoding cluster for both comparisons. An off-diagonal element was deemed dynamic if both tests were significant (p<0.05), and both corresponding diagonal elements were part of the above-chance decoding cluster.

Following *Spaak et al., 2017*, we also computed the dynamicism index as a proportion of elements across time that were dynamic. Specifically, we calculated the proportion of (off-diagonal) dynamic elements corresponding to a diagonal time point in both columns (corresponding to the test time points) and rows (corresponding to the train time points) of the temporal cross-decoding matrix.

## Temporal cross-decoding simulations

To address the possibility that the dynamic clusters in the temporal cross-decoding analysis might arise as a result of the decoder picking up on different features within the signal as a function of the SNR, we ran three temporal cross-decoding simulations where the train and test data had varying levels of SNR.

In the first simulation, we created a dataset of 200 voxels that had a sine or cosine response function to orientations between 1° and 180°, the same orientations as the remembered target. A circular shift was applied to each voxel to vary preferred (or maximal) responses of each simulated voxel. This resulted in a dataset that captured the neural population's response to all orientations. We then assessed the decoding performance under different SNR conditions during training and testing. In total, we ran seven iterations of the simulation, which correspond to the number of subjects in the second experiment. For each iteration, we randomly selected 108 responses from the full set of 180 for training, and then independently sampled another 108 from the same full set for testing. This ensured that the same orientation could appear in both sets, consistent with the structure of the original experiment. To increase variability, the selected trials differed in each iteration. Random white noise was applied to the data, and thus the SNR was independently scaled according to the specified levels for train and test data. We then use the same pSVR decoder as in the temporal cross-decoding analysis to train and test. We plot the SNR cross-decoding matrix as temporal decoding matrices (*Figure 1—figure supplement 2a*).

The second and third simulations were conducted to investigate whether increased noise levels would induce the decoder to rely on different features of the no-distractor and noise distractor data from the first experiment. We used empirical data from the primary visual cortex (V1) under the no-distractor and noise distractor conditions for the second and third simulations, respectively. Data from time points 5.6–8.8 s after stimulus onset (*Figure 1c*, V1) were averaged across five TRs. As in the first simulation, SNR was systematically manipulated by adding white noise. Additionally, to see whether

the initial decrease in SNR and subsequent increase (as seen first during the delay and increase during the response period) would result in dynamic coding clusters, we initially increased and subsequently decreased the amplitude of added noise. The same pSVR decoder was used to train and test on the data in a cross-validated fashion with different levels of added noise. We plot the cross-decoding matrix as in the first simulation (*Figure 1—figure supplement 2b and c*).

## Neural subspaces

We adapted the method from *Li and Curtis, 2023*, to calculate two-dimensional neural subspaces encoding VWM information at a given time point. To do so, we used PCA. To maximize power, we binned trial-wise fMRI activations into four equidistant bins of 45° and averaged the signal across all trials within a bin (*Figure 2a*). The data matrix $\mathbf{X}$ was defined as a $\rho \times \upsilon$ matrix, where $\rho=4$ was the four orientation bins, and $\upsilon$ was the number of voxels. We mean-centered the columns (i.e. each voxel) of the data matrix.

This analysis focused on the time points from 4 s to 17.6 s after delay onset. The first TRs were not used since the temporal cross-decoding results showed no above-chance decoding. We averaged across every three TRs leading to six nonoverlapping temporal bins resulting in six $\mathbf{X}$ matrices. We calculated the principal components using eigen decomposition of the covariance matrix for each $\mathbf{X}$ and defined the matrix $\mathbf{V}$ using the two largest eigenvalues as a $\upsilon \times 2$ matrix, resulting in six neural subspaces, one for each nonoverlapping temporal bin.

## Neural subspaces across time

For visualization purposes, we used three out of the total of six neural subspaces from the following time points: early (7.2 s), middle (12 s), and late (16.8 s). Following the aforementioned procedure, these subspaces were calculated on half of the trials, as we projected the left-out data onto the subspaces. The left-out data were binned into six temporal bins between 4 s and 17.6 s after target onset with no overlap, just like in the calculation of the six subspaces. The projection resulted in a $\rho \times 2$ matrix $\mathbf{P}$ for each projected time bin (resulting in a total of six $\mathbf{P}$ matrices). We use distinct colors to plot the temporal trajectories of each orientation bin across time in a 2D subspace flattened (*Figure 2c*) and not flattened (*Figure 2—figure supplement 1*) across the time dimension. Importantly, the visualization analysis was done on a combined participant-aggregated V1-V3AB region, which included all reliable voxels across the four regions and all six participants (see *Voxel selection*).

To measure the alignment between coding spaces at different times, we calculated an aPA between all subspaces (*Figure 2c*). We used the MATLAB function `subspace` for an implementation of the method proposed by *Björck and Golub, 1973*, to measure the angle between two $\mathbf{V}$ matrices. This provided us with a possible principal angle between 0° and 90°; the higher the angle, the larger the difference between the two subspaces. To avoid overfitting and as in the visualization analysis, we used a split-half approach to compute the aPA between subspaces. Half of the binned trials were used to calculate $\mathbf{V}_{i,A}$ and $\mathbf{V}_{j,A}$, and half for $\mathbf{V}_{i,B}$ $\mathbf{V}_{j,B}$, where $\mathbf{A}$ and $\mathbf{B}$ refer to the two halves of the split, and $\mathbf{i}$ and $\mathbf{j}$ refer to the two time bins compared. For significance testing, the within-subspace angle (the angle between two splits of the data within a given temporal bin [i.e. $\mathbf{V}_{i,A}$ and $\mathbf{V}_{i,B}$]) was subtracted from the between-subspace PA (the angle between two different temporal bins [e.g. $\mathbf{V}_{i,A}$ and $\mathbf{V}_{j,B}$]). Unlike the visualization analysis, the PA was calculated per participant 1000 times using different splits of the data on a combined V1-V3AB region that included the reliable voxels across the four regions (see *Voxel selection*). The final aPA value was an average across all iterations for each participant.

## Sensory distractor and orientation VWM target neural subspaces

For the orientation VWM target and sensory distractor neural subspace, we followed the aforementioned subspace analysis, but instead of calculating subspaces on six temporal bins, we averaged across the 4–17.6 s delay period and calculated a single subspace. As in the previous analysis, we split the orientation VWM trials in half. We then binned the trials either based on the target orientation or the sensory distractor. For visualization purposes, we projected the left-out data averaged based on the sensory distractor and the target onto subspaces derived from both the sensory distractor and target subspaces. As in the previous visualization, the analysis was run on a participant-aggregated V1-V3AB region.

To calculate the aPA, we had the following subspaces: $V_{Target,A}$, $V_{Dist,A}$, $V_{Target,B}$, and $V_{Dist,B}$, where the subspaces were calculated on trials binned either based on the target orientation or the sensory distractor. The aPA was calculated by subtracting the within-subspace angle ($V_{Target,A}$ and $V_{Target,B}$, $V_{Dist,A}$ and $V_{Dist,B}$) from the sensory distractor and working memory angle ($V_{Target,A}$ and $V_{Dist,B}$, $V_{Target,B}$ and $V_{Dist,A}$). The split-half aPA analysis was performed 1000 times, and the final value was an average across these iterations for each participant.

## Acknowledgements

JKD was funded by the Max Planck Society and BMBF (as part of the Max Planck School of Cognition). JDH was supported by the Deutsche Forschungsgemeinschaft (DFG, Exzellenzcluster Science of Intelligence); SFB 940 'Volition and Cognitive Control'; and SFB-TRR 295 'Retuning dynamic motor network disorders using neuromodulation'. SW was supported by Deutsche Forschungsgemeinschaft (DFG) Research Training Group 2386 451 and EXC 2002/1 'Science of Intelligence'. We thank Rosanne Rademaker, Chaipat Chunharas, and John Serences for collecting and sharing their data open access, without which this reanalysis would not have been possible. We also thank Rosanne Rademaker, Michael Wolff, Amir Rawal, and Maria Servetnik for extensive discussions of the results. We also thank Vivien Chopurian and Thomas Christophel for their feedback on the manuscript.

## Additional information

### Funding

| Funder | Grant reference number | Author |
|---|---|---|
| Max Planck School of Cognition | | Jonas Karolis Degutis |
| Deutsche Forschungsgemeinschaft | SFB 940 | John-Dylan Haynes |
| Deutsche Forschungsgemeinschaft | Research Training Group 2386 451 | Simon Weber |
| Deutsche Forschungsgemeinschaft | SFB-TRR 295 | John-Dylan Haynes |
| Deutsche Forschungsgemeinschaft | EXC 2002/1 | Simon Weber |

The funders had no role in study design, data collection and interpretation, or the decision to submit the work for publication.

### Author contributions

Jonas Karolis Degutis, Conceptualization, Data curation, Software, Formal analysis, Funding acquisition, Validation, Investigation, Visualization, Methodology, Writing – original draft, Writing – review and editing; Simon Weber, Joram Soch, Software, Methodology, Writing – review and editing; John-Dylan Haynes, Supervision, Writing – review and editing, Funding acquisition

### Author ORCIDs

Jonas Karolis Degutis ⓘ https://orcid.org/0009-0003-8682-3920
Simon Weber ⓘ https://orcid.org/0000-0002-8440-1156
Joram Soch ⓘ http://orcid.org/0000-0002-8879-5666
John-Dylan Haynes ⓘ http://orcid.org/0000-0003-1786-6954

### Ethics

The study's dataset (Rademaker et al. 2019) was acquired at the University of California, San Diego, and was approved by the local Institutional Review Board. The participants provided written informed consent and were monetarily reimbursed.

Reviewer #1 (Public review): https://doi.org/10.7554/eLife.99290.4.sa1

Author response https://doi.org/10.7554/eLife.99290.4.sa2

---

## Additional files

### Supplementary files
Supplementary file 1. Supplementary Tables.

MDAR checklist

### Data availability
The preprocessed data are shared open-access at https://osf.io/dkx6y/. The analysis scripts and results are shared at https://github.com/degutis/WM_dynamicCoding (copy archived at *Degutis, 2025*) and https://osf.io/jq3ma/, respectively.

The following dataset was generated:

| Author(s) | Year | Dataset title | Dataset URL | Database and Identifier |
|---|---|---|---|---|
| Degutis JK | 2024 | Neural dynamics of visual working memory representation during sensory distraction | https://osf.io/jq3ma/ | Open Science Framework, jq3ma |

The following previously published dataset was used:

| Author(s) | Year | Dataset title | Dataset URL | Database and Identifier |
|---|---|---|---|---|
| Rademaker RL, Chunharas C, Serences JT | 2019 | Coexisting representations of sensory and mnemonic information in human visual cortex | https://osf.io/dkx6y/ | Open Science Framework, dkx6y |

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
