## [Editor Report · eLife Assessment]

This **important** study reports a reanalysis of one experiment of a previously-published report to characterize the dynamics of neural population codes during visual working memory in the presence of distracting information. This paper presents **solid** evidence that working memory representations are dynamic and distinct from sensory representations of intervening distractions. This research will be of interest to cognitive neuroscientists working on the neural bases of visual perception and memory.

---

## [Referee Report · Reviewer #1 (Public review)]

Summary:

In this study, the authors re-analyzed a public dataset (Rademaker et al, 2019, Nature Neuroscience) which includes fMRI and behavioral data recorded while participants held an oriented grating in visual working memory (WM) and performed a delayed recall task at the end of an extended delay period. In that experiment, participants were pre-cued on each trial as to whether there would be a distracting visual stimulus presented during the delay period (filtered noise or randomly-oriented grating). In this manuscript, the authors focused on identifying whether the neural code in retinotopic cortex for remembered orientation was 'stable' over the delay period, such that the format of the code remained the same, or whether the code was dynamic, such that information was present, but encoded in an alternative format. They identify some timepoints - especially towards the beginning/end of the delay - where the multivariate activation pattern fails to generalize to other timepoints, and interpret this as evidence for a dynamic code. Additionally, the authors compare the representational format of remembered orientation in the presence vs absence of a distracting stimulus, averaged over the delay period. This analysis suggested a 'rotation' of the representational subspace between distracting orientations and remembered orientations, which may help preserve simultaneous representations of both remembered and viewed stimuli. Intriguingly, this rotation was a bit smaller for Expt 2, in which the orientation distractor had a greater behavioral impact on the participants' behavioral working memory recall performance, suggesting that more separation between subspaces is critical for preserving intact working memory representations.

Strengths:

(1) Direct comparisons of coding subspaces/manifolds between timepoints, task conditions, and experiments is an innovative and useful approach for understanding how neural representations are transformed to support cognition

(2) Re-use of existing dataset substantially goes beyond the authors' previous findings by comparing geometry of representational spaces between conditions and timepoints, and by looking explicitly for dynamic neural representations

(3) Simulations testing whether dynamic codes can be explained purely by changes in data SNR are an important contribution, as this rules out a category of explanations for the dynamic coding results observed

Weaknesses:

(1) Primary evidence for 'dynamic coding', especially in early visual cortex, appears to be related to the transition between encoding/maintenance and maintenance/recall, but the delay period representations seem overall stable, consistent with some previous findings. However, given the simulation results, the general result that representations may change in their format appears solid, though the contribution of different trial phases remains important for considering the overall result.

(2) Converting a continuous decoding metric (angular error) to "% decoding accuracy" serves to obfuscate the units of the actual results. Decoding precision (e.g., sd of decoding error histogram) would be more interpretable and better related to both the previous study and behavioral measures of WM performance.

Comments on revised version:

The authors have addressed all my previous concerns.

---

## [Author Response]

The following is the authors’ response to the previous reviews

**Reviewer #1:**

**Reviewer #1 (Recommendations For The Authors):**
(1) At several places in the reply to reviewers and the manuscript, when discussing the new simulations conducted, the authors mention they break the 180 trials into a train/test split of 108/108 - is this value correct? If so, how? (pg 19 of updated manuscript)

Thank you for pointing this out; it was not clearly explained. We have now added the explanation to the Methods section:

“For each iteration, we randomly selected 108 responses from the full set of 180 for training, and then independently sampled another 108 from the same full set for testing. This ensured that the same orientation could appear in both sets, consistent with the structure of the original experiment.”

(2) I appreciate the authors have added the variance explained of principal components to the axes of Fig. 3, though it took me a while to notice this, and this isn't described in the figure caption at all. It would likely help readers to directly explain what the % means on each axis of Fig. 3.

Thank you, we have now added a description in both Fig. 2 and 3:

“The axes represent the first two principal components, with labels indicating the percent of total explained variance.”

(3) I believe there is a typo/missing word in the new paragraph on pg 15: "neural visual WM representations in the early visual cortices are [[biased]] towards distractors" (I think the bracketed word may be omitted as a typo)

Thank you - fixed.